# Investigation of the Effect of Written and Visual Information on Anxiety Measured Before Magnetic Resonance Imaging: Which Method is Most Effective?

**DOI:** 10.3390/medicina56030136

**Published:** 2020-03-18

**Authors:** Burkay Yakar, Edibe Pirinçci

**Affiliations:** 1Department of Family Medicine, Faculty of Medicine, Firat University, 23119 Elazig, Turkey; 2Department of Public Health, Faculty of Medicine, Firat University, 23119 Elazig, Turkey; edibepirinci@yahoo.com

**Keywords:** magnetic resonance imaging, anxiety, information, STAI, MRI

## Abstract

*Background and objective:* Magnetic resonance imaging (MRI) causes severe anxiety in some patients. Anxiety during MRI leads to prolongation of the procedure and deterioration of image quality, resulting in loss of labor and cost increase. The aim of this study was to investigate the effect of written and visual information on state anxiety in patients undergoing MRI. *Material and Methods:* A cross-sectional prospective study was conducted with 294 participants. The study was carried out between January 2019 and March 2019 at the Radiology Clinic of the tertiary university hospital. The participants were divided into 3 groups as group 1 (control group), group 2 (written information) and group 3 (visual information). The trait anxiety and state anxiety of the participants were measured by State-Trait Anxiety Inventory (STAI) inventory, which can measure both anxiety status. *Results:* There was no statistically significant difference between demographic characteristics and trait anxiety scores (*p* = 0.20) of all three groups. The state anxiety scores of group 3 were statistically lower than the group 2 (*p* < 0.001) and control group (*p* < 0.001). The state anxiety scores of group 2 were statistically lower than control group (*p* < 0.001). *Conclusion:* MRI anxiety can be reduced by visual and written information. Visual information may be more effective in reducing MRI anxiety than written information.

## 1. Introduction

Anxiety is an important health problem due to its high prevalence today [1]. Anxiety can cause intense stress, loss of attention and concentration, learning difficulties, increased probability of errors in work and actions, deterioration in human relations and decreased work efficiency [2]. In addition to these negative effects of anxiety on daily routines of people, it also affects human health negatively by paving the way for many diseases such as coronary problems [3]. For this reason, healthcare professionals should protect patients from anxiety and stress as much as possible.Research shows that the prevalence of anxiety disorder or anxiety symptoms in the society is between 10–70% [4]. It has been reported that being a patient, being admitted to hospital and being exposed to a procedure for diagnosis and treatment may cause stress andnegative effects such as anger, fear, anxiety. Patients may feel insecure and experience anxiety as a result of the equipment, smell, sound and procedures of which they are not aware, which is a different environment for them [5]. Another factor that causes anxiety in patients is MRI procedures. The loud noise, fear of pain, the experience of losing control, the long duration of the examination andenclosed environment of the MR tunnel is known to induce anxiety in patients [6].

Magnetic resonance imaging (MRI) has become one of the most important diagnostic tools in many areas of medicine. Several studies have shown that MRI causes severe anxiety in 37% of patients, even those who state that they are not claustrophobic, and 5–10% cannot complete MRI due to claustrophobia [7]. Patients cannot provide effective cooperation since they feel “buried alive” “abandoned” and consequently [8], movement-induced artifacts cause deterioration of image quality and prolongation of processing time [7,9]. Long-term screening and additional sessions require sedation or general anesthesia in claustrophobic patients, which increases screening costs. In order to reduce anxiety during MRI, many approaches such as sedation, lying prone instead of supine, rehearsal with MRI simulator before MRI, relaxation exercises and information are recommended [10,11]. Although all these approaches are valuable, sedation and lying prone may not be suitable for brain or spinal imaging [11].

The aim of this study was to investigate the effect of written and visual information on state anxiety in patients undergoing MRI.

## 2. Materials and Methods

### 2.1. Study Design and Population

This study was designed as a cross-sectional prospective. The study was carried out between January 2019 and March 2019 at the Radiology Clinic of the tertiary university hospital. Ethical approval was obtained from the Ethics Committee of Non-Interventional Research of Firat University (31.05.2018/10.14). Those who met the inclusion criteria were included in the study and written consent form was obtained from all participants.

The patients who underwent MRI for the first time were included in the study. The study inclusion criteria were as follows: (i) undergoing MRI for the first time (ii) being over 18 years old and (iii) undergoing thoracic or abdominal MR imaging. On the other hand, the exclusion criteria were: (i) having psychiatric and neurological diseases that affect cognitive functions, (ii) having anxiety disorder, (iii) sedative drug or substance use, (iv) having claustrophobia,(v) illiteracy and (vi) undergoing MRI before.

### 2.2. Sample Size and Selection

The population of the study consisted of patients appointed for MRI in the university hospital. The sample size that should be included in the study was calculated with the formula (n = t^2^ × p × q/d^2^). Since the prevalence of anxiety was evaluated as 30% during MRI, the size of the sample was calculated as 324 participants.The study was planned with 3 groups of participants. Study groups were designed as group 1 (control group), group 2 (Group with written information about MRI) and group 3 (Group with visual information about MRI). Considering the size of the sample, each group was planned to have at least 100 participants. As the routine procedure of the hospital where the study was conducted, the patients examined in different clinics and requested MRI are given an appointment for a future date and time. Patients are asked to be in the radiology clinic at least 30 min before the MRI appointment time. Participants who were given the appointment date and applied for MRI were included in the study. No participants without an appointment were included in the study. As a result, we reached 100 patients in each group and a total of 300 participants. Participants were randomly divided into the groups. The participants in group 1 were informed about the study and asked to fill the questionnaire forms (demographics form, State Anxiety Inventory (SAI) and Trait Anxiety Inventory (TAI)). Written information brochures on MRI were prepared for the participants in group 2. After the participants were first informed about the study, written brochures were distributed to all participants in group 2. Participants were asked to read the brochures. After the participants read the brochures, all participants were asked to fill all questionnaire forms (socio-demographic, SAI and TAI forms) used in the study. Visual information videos with the same content as written information form were prepared for the participants in group 3. Participants in group 3 were informed about the study and then all participants watched a video on tablet computers. After the end of the video, questionnaire forms (socio-demographic, SAI and TAI forms) were applied to the participants. At the end of data collection, six participants who completed the questionnaire forms incompletely or incorrectly were excluded from the study. Finally, 294 participants were enrolled as Control group (group 1, *n* = 97), group 2 (Group with written information about MRI, *n* = 100) and group 3 (Group with visual information about MRI, *n* = 97). All these procedures were performed in the waiting room before MRI.

### 2.3. Data Collection Tools

Research data were collected by using a structured questionnaire developed as a result of review of literature (demographics form) and State-Trait Anxiety Inventory (STAI) questionnaire. All participants were asked to fill out a demographics form (consisting of sex, age, education, marital status) and State-Trait Anxiety Inventory (STAI) forms. STAI is developed by Spielberger, Gorsuch and Lushene in English. It has 2 subdivisions: State Anxiety Inventory (SAI) which measures the anxiety of a cross-section and Trait Anxiety Inventory (TAI) which measures the susceptibility to anxiety. Range of each subdivision’s score can be between 20 and 80 and higher scores mean higher anxiety. The STAI form was applied to all participants in the local language which is Turkish. STAI has shown validity and reliability in its original language and in Turkish [12,13].

### 2.4. Information Method

Routine information procedure of the radiology clinic prior to MRI was made by clinical staff to all patients. Written brochures and visual videos were prepared to inform the participants except for the routine procedure of the radiology clinic. The informative content of the written information brochure and the visual information video were similar. In both cases, it included information such as MRI, rules to be considered, how long it lasts, what do staff do during the MR imaging process, whether there is pain or if there is a sound. Information was given by the same researcher during the study, and consisted of standard messages in both of the information methods:MRI is a safe and painless method even if they are in a closed tunnel.The officials are watching the patients from outside.You will hear sound during the procedure, but this sound can be heard routinely during MRI (This is the usual situation and is not dangerous).MRI will take about 20 min.

All participants in the study groups (groups 2 and 3) were informed about this standard information. The visual information video was about 5 min long. Visual video was recorded by the researchers. The same information used in the written brochure was explained by a technician in the visual video. In the video, the participants were shown the MRI device, the sound and the magnetic resonance imaging.

In all three groups, MRI was performed within 20 min after filling STAI forms. All this procedure was performed in the MRI waiting room. MRI of all participants was performed with the same MR imaging device. For imaging purposes, a 1.5 Tesla, 60 cm gantry opening, Philips Intera MRI device (Philips Medical Systems, Amsterdam, The Netherlands) was used.

### 2.5. Statistical Analysis

Statistical analysis of the data was performed by IBM SPSS 22 statistics package program. Shapiro-Wilk test was used to determine whether the data showed normal distribution. Descriptive statistics of the data were expressed as mean ± standard deviation for variables with normal distribution in continuous data, (median (minimum: maximum)) for non-normal distribution variables, and frequency for categorical variables as percentage (*n* (%)). In comparison of more than two independent groups, One-Way ANOVA and LSD test for post-Hoc test were used for normal distributed continuous data, Kruskal Wallis test and Dunn test for post-Hoc test for non-normal distributed continuous data. Pearson chi-square test was used to analyze categorical data. Significance level was α = 0.05. Statistically significant significance values are indicated in bold in the tables.

## 3. Results

There were 294 patients enrolled in the study. One hundred of them in the written informed group (group 2), 97 of them in the visual informed group (group 3) and 97 were control patients (group 1). The mean age of the participants included in the study was 42.99 ± 3.44 years. 50.7% (*n* = 149) of the participants were female and 49.3% (*n* = 49.3) were male. When the 3 groups were compared in terms of age, gender, marital status, education level, and chronic disease status, there was no statistically significant difference between demographic characteristics of 3 groups (Table 1).

In the study, firstly, the trait anxiety states of all participants were measured. The trait anxiety scores of the participants were group 1: 42.63 ± 7.87, group 2: 44.46 ± 6.89 and group 3: 44.07 ± 7.70 respectively. There was no statistically significant difference was found between the trait anxiety scores of the participants in all 3 groups (*p* = 0.20) (Table 2).

State anxiety scores of all participants were measured after written or visual information in the case group and without given information in the control group. The state anxiety scores of the participants were group 1: 56.00 (30.0–74.0), group 2: 44.00 (30.0–62.0) and group 3: 34.00 (20.0–60.0) respectively. There was a statistically significant difference between state anxiety score of three groups of participants (*p* < 0.001). The state anxiety scores of the participants who were given visual and written information were lower than those who were not informed (*p* < 0.001). The state anxiety scores of the participants who were given visual information were lower than the anxiety scores of the participants who were given written information (*p* < 0.001) (Table 3).

## 4. Discussion

This study investigated the effect of written and visual information on anxiety before MRI. Anxiety levels of participants during MRI were measured by State Anxiety Inventory (SAI). Information was given to the both study groups’ participants as standard messages. No information was given to the participants in the control group.

Earlier studies have shown that MRI and BT tests cause anxiety in patients [5,14]. Many clinicians agree that anxiety causes movement artifact and incomplete processing during the MRI [15,16,17]. Prolonged and repeated MRI procedure result in decrease in the diagnostic value of MRI and deterioration of image quality. As a result, artifacts will limit the performance of MRI, leading to the loss of valuable time for personnel and equipment, leading to increased costs [18]. In this context, we aimed to reduce the anxiety during MRI with our interventions.

The current study showed that both written and visual information reduced state anxiety during the MRI. Gray et al. sent information brochures to the patients who would undergo MRI and it was found that the anxiety status of the group was decreased [11]. Similarly, in two different studies in the literature, written information before gastroscopy has been shown to reduce patients’ anxiety [19,20]. In our study, the anxiety scores of the group informed by written brochures were found to be lower than the control group. In this sense, our findings support the data in the literature. Therefore, patients’ status anxiety can be reduced with written information before MRI.

Another aim of our study was to investigate the effect of visual information on anxiety. Two studies investigated the effect of visual information on anxiety during MRI and reported that visual information reduced anxiety [21,22]. Acay et al. reported that the SAI scores of the group with visual information were significantly lower than the control group [21]. Tazegul et al. measured both SAI and cortisol levels of participants to investigate the effect of visual information on anxiety during MRI. Tazegul et al. reported that both SAI scores and cortisol levels were significantly lower in the visual information group compared to the control group [22]. Similarly, in our study, the anxiety score of the group with visual information was found to be statistically lower than the control group. Literature data and our findings have shown that informing patients before MRI may reduce state anxiety. In previous studies, the effect of individual written and visual information on anxiety during MRI was emphasized. In the present study, state anxiety scores of the group with visual information were found to be statistically lower than the group with written information. In addition to the data in the literature, we can say that visual information is more effective in reducing state anxiety before MRI. Our study may be the first study in which both written and visual information is applied together.

### Limitations

Although the 3 groups are homogeneous, people’s previous health experiences may affect anxiety. In the present study, past illness and hospital experiences of the participants were not questioned. Anxiety states were assessed by the participants’ own statements. The use of biophyschometric scale and biochemical markers in the assessment of anxiety could have provided more effective data. Another limitation of the study is the evaluation of patients’ anxiety status before MRI. Although given information was found to reduce anxiety, the anxiety status of patients could not be evaluated during MRI.

Another limitation of the article is that although the content of written and visual information is similar, visual information differs from written information due to its audio and video content. We suggest that further studies should make the two conditions more similar. Despite our limitations, conducting the study with a large population increases the importance of our study. Our study included one of the largest populations in the literature. We believe that this makes our findings valuable. Another important aspect of our study is the simultaneous application of written and visual information.

In similar studies that will be done in the future, the anxiety status of the participants can be examined with biopyscometric and biochemical methods. In the current study, patients were informed individually with tablet computers. It is obvious that such information is not an easy method. The effect of visual information on anxiety can be investigated by watching information videos from televisions to in waiting rooms. In this way practical and cheaper patient information can be achieved.

## 5. Conclusions

In conclusion, anxiety during MRI can be reduced by both written and visual information. Visual information was found to be more effective in reducing anxiety before MRI than written information. By applying cheap and easy information methods, we can perform more effective MRI so that we can reduce both labor loss and costs.

## Figures and Tables

**Table 1 medicina-56-00136-t001:** Sociodemographic characteristics of patients.

Variables		Group 1*n* (%)		Group 2*n*(%)		Group 3*n* (%)		Total *N* (%)		*p* Value
Age [Median(Min:Max)]		40.00 (21–75)		41.00 (18–70)		40.00 (18–75)		42.00 (18–75)		KW: *p* = 0.52 *
Gender	Female	50	(33.6)	49	(32.9)	50	(33.6)	149	(50.7)	χ2: 0.17 *p* = 0,92 **
	Male	47	(32.4)	51	(35.2)	47	(32.4)	145	(49.3)	
Marital Status	Married	83	(35.0)	83	(35.0)	71	(30.0)	237	(80.6)	χ2: 5.30 *p* = 0.07 **
	Single/divorced	14	(24.6)	17	(29.8)	26	(45.6)	57	(19.4)	
Income level perception	İnsufficient	12	(42.9)	5	(17.9)	11	(39.3)	28	(9.5)	χ2: 6.37 *p* = 0.17 **
	Sufficient	79	(31.0)	92	(36.1)	84	(32.9)	255	(86.7)	
	Good	6	(54.5)	3	(27.3)	2	(18.2)	11	(3.7)	
Education	Elementary school	44	(35.2)	44	(35.2)	37	(29.6)	125	(42.5)	χ2: 5.08 *p* = 0.75 **
	High school	53	(31.4)	56	(33.1)	60	(35.5)	95	(32.3)	
	College	28		23		23		74	(25.2)	
Chronic disease	Have	40	(33.9)	40	(33.9)	38	(32.2)	118	(40.1)	χ2:0.09 *p* = 0.96 **
	Have not	57	(32.4)	60	(34.1)	59	(33.5)	176	(59.9)	

*n*: number, %: percent, * Kruskal Wallis test statistic (KW), **Pearson Chi-square test statistic (χ2).

**Table 2 medicina-56-00136-t002:** Trait anxiety scores of participants.

Groups	Trait Anx.	Score Mean Std	*p* Value *	*p* Value **
Group 1 (*n* = 97)	42.63	±7.87	F = 1.61 *p* = 0.20 ***	F = 1.071–2: 0.09 ***
Group 2 (*n* = 100)	44.46	±6.96		F = 1.081–3: 0.18 ***
Group 3 (*n* = 97)	44.07	±7.70		F = 1.072–3: 0.72 ***

* multiple comparison *p* value, ** binary comparison *p* value, *** One-Way ANOVA test statistic.

**Table 3 medicina-56-00136-t003:** State anxiety scores of participants.

Groups	State Anxiety Scores	*p* Value *	*p* Value **
	Median (min–max)		
Group 1	56.00 (30.0–74.0)	KW:105.27 *p* <0.001 ***	KW:6.021–2: <0.001 ***
Group 2	44.00 (30.0–62.0)		KW: 12.181–3: <0.001 ***
Group 3	34.00 (20.0–60.0)		KW:5.792–3: <0.001 ***

* multiple comparison *p* value, ** binary comparison *p* value, *** Kruskal-Wallis test statistic.

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
