# Peer review of "Investigation of the Effect of Written and Visual Information on Anxiety Measured Before Magnetic Resonance Imaging: Which Method is Most Effective?"

_medicina, 2020, doi:10.3390/medicina56030136_

Round 1

Reviewer 1 Report

Thanks for letting me read this manuscript.

Title

Perhaps: Investigation of the effect of written and visual information on anxiety measured before magnetic resonance imaging: Which method is most effective?

Abstract

Line 10, causes severe anxiety in patients. MRI cause not severe anxiety in all patients but in some.

Introduction

Line 28-34: this is not relevant for the description of anxiety and claustrophobia in relation to MRI.

Why are MRI-examinations causing anxiety? Shortly describe.

Line 36: the statement that 10-70% in the society has symptoms of anxiety disorders has a reference, still there is a large difference between 10 and 70%. Is this correct?

Line 42: Several studies have shown that MRI causes severe anxiety in 37% of patients and 5-10% cannot complete MRI due to claustrophobia. Those statements need references.

Line 44-45: Are “be buried alive” and “to be abandoned” citations? If so, the reference shall be written directly after.

Line 47:  It should probably be “sedation OR anesthesia”.

Line 50-51: Lying on face is not proper language in an academic publication. Use prone and supine.

Material and method

Line 61: Patients are not included in the study before they have given written informed consent. They are asked to participate.

Line 66-68: You are just calculating the sample size. Don’t use universe, even if it is understandable.

Line 73-88: How were the patients divided into the three groups? Randomized? In a special order?

When were the patients invited to participate? By letter? When they arrived to the MRI-unit?

Were patients referred to all kind of MRI-examinations asked to participate, or just some kind?

Line 103: shooting is not an academic language.

Line 117: Here it is written that the participants answer the questionnaire about anxiety before examination. If that is correct the anxiety is not measured during MRI which is written in the manuscript.

The material and method part of the manuscript needs be rewritten in a more structured way. Specially “study design and population”.

Result

The result part is defective. With 97-100 participants in every group it is possible to calculate differences between gender and ages (young and old).  This will also make it easier to write the discussion part, which is very short.

In all tables it shall be written how the p-value is calculated. It is not enough to have it in the statistical analysis part. All abbreviations shall be explained even “n”.

There are many tables and very little text in the result.

Discussion

Line 153: It is written that the anxiety during MRI were measured, this is not correct. It was measured before MRI.

Line 154: Were no information at all given to the control group?

Line 158: shots are not academic language.

Gray et al is referred to as a study evaluating the effect of written information prior MRI for reducing patients anxiety. It is not clear if you refer to the anxiety measured before or after MRI-examinations. There are several studies evaluating patient’s anxiety during MRI after written or visual information given before the examination. Perhaps your literature review is insufficient.

Several studies have evaluated patient anxiety during MRI examinations after both visual and written information. In the present study patient anxiety is measured before MRI and that is not well described in the literature. But this has to be clarified in the manuscript. In the discussion it is written that anxiety was measured during MRI and that is not the case. It will increase the value of your work if it is pointed out that you measure the anxiety before examination.

All abbreviations are not explained.

Table references in the text shall be written before the dot.

Author Response

dear reviewer;

all revision suggestions have been applied

revisions are given in the attachment

-----

Response to Reviewer 1 Comments

Point 1: Title:  Perhaps: Investigation of the effect of written and visual information on anxiety measured before magnetic resonance imaging: Which method is most effective?

Response 1: Investigation of the effect of written and visual information on anxiety measured before magnetic resonance imaging: Which method is most effective? (in box WK 1)

Point 2: Line 10, causes severe anxiety in patients. MRI cause not severe anxiety in all patients but in some.

Response 2: Magnetic resonance imaging (MRI) causes severe anxiety in some patients. (in box WK 2)

Point 3: Introduction:   Line 28-34: this is not relevant for the description of anxiety and claustrophobia in relation to MRI.

Why are MRI-examinations causing anxiety? Shortly describe. ( Line 49 )

Line 36: the statement that 10-70% in the society has symptoms of anxiety disorders has a reference, still there is a large difference between 10 and 70%. Is this correct?  (line 45: literature reference is provided for this information.)

Line 42: Several studies have shown that MRI causes severe anxiety in 37% of patients and 5-10% cannot complete MRI due to claustrophobia. Those statements need references. (line 59 : reference 7)

Line 44-45: Are “be buried alive” and “to be abandoned” citations? If so, the reference shall be written directly after. (line 61:  reference 8)

Line 47:  It should probably be “sedation OR anesthesia”.  (line 67)

Line 50-51: Lying on face is not proper language in an academic publication. Use prone and supine.  (line 69)

Response 3: all these revision suggestions have been applied in box WK3, WK 4, WK5, WK6, WK7, WK8 AND WK 9)

Point 4: Material and method

Line 61: Patients are not included in the study before they have given written informed consent. They are asked to participate. (line 80-81)

Line 66-68: You are just calculating the sample size. Don’t use universe, even if it is understandable. (line 91-92)

Line 73-88: How were the patients divided into the three groups? Randomized? In a special order? (line 105-107)

When were the patients invited to participate? By letter? When they arrived to the MRI-unit?  (line 97-105)

Were patients referred to all kind of MRI-examinations asked to participate, or just some kind? (line 82-85)

Line 103: shooting is not an academic language. (line 140-141)

Line 117: Here it is written that the participants answer the questionnaire about anxiety beforeexamination. If that is correct the anxiety is not measured during MRI which is written in the manuscript.  (line 100)

 The material and method part of the manuscript needs be rewritten in a more structured way. Specially “study design and population”. (2.1. Study design and population, 2.2. Sample size and selection 2.3. Data collection tools  2.4. Information method  2.5. Statistical analysis)

Response 4: all these revision suggestions have been applied in box WK11, WK 12, WK13, WK15,)

Point 5: Results:  In all tables it shall be written how the p-value is calculated. It is not enough to have it in the statistical analysis part. All abbreviations shall be explained even “n”. (table 1 table 2 and table 3: statistical methods and abbreviations added)

There are many tables and very little text in the result. (line 173-178 and line 183-186 and line: 192-199)

Point 6: Discussion

Line 153: It is written that the anxiety during MRI were measured, this is not correct. It was measured before MRI. (line 208)

Line 154: Were no information at all given to the control group? (line 210-211)

Line 158: shots are not academic language. (line 214-216)

Gray et al is referred to as a study evaluating the effect of written information prior MRI for reducing patients anxiety. It is not clear if you refer to the anxiety measured before or after MRI-examinations. There are several studies evaluating patient’s anxiety during MRI after written or visual information given before the examination. Perhaps your literature review is insufficient. (line:220-227)

Several studies have evaluated patient anxiety during MRI examinations after both visual and written information. In the present study patient anxiety is measured before MRI and that is not well described in the literature. But this has to be clarified in the manuscript. In the discussion it is written that anxiety was measured during MRI and that is not the case. It will increase the value of your work if it is pointed out that you measure the anxiety before examination. (line: 212-268)

Response 6: all these revision suggestions have been applied in box WK2, WK 22, WK3)

Reviewer 2 Report

This paper aims to investigate the effect on anxiety state of written vs. visual information in patients undergoing MRI. There are some recent, but scarce studies evaluating the effects of different  types of communication (written, visual and by phone), but most of them compare only one type of communication to a control group. Thus, the main contribution of this paper is to compare simultaneously three different conditions to communicate MRI information to patients and its effects on state anxiety prior to MRI.

The study includes a large sample of patients, which is a strength, since carrying out studies in real work contexts its methodologically challenging and sometimes difficult.

On another hand, some conceptual questions and inferences from results should be object of concern and reformulated. Specific comments on these issues are mentioned ahead:

  • In Introduction, sentence from line 46 to 50 about strategies to reduce patients anxiety should be supported by references, highlighting the importance of MRI previous information in anxiety, to support the present study (e.g. https://doi.org/10.1016/j.pec.2019.08.003, or other)
  • In study design, a major issue which is not explained and may raise ethical questions is the control group. This group, according to good practices standards in MRI, should receive some information about the procedure before it starts. Usually, a technician or an assistant give some verbal information to the patient before the procedure (welcomes him, shows him the changing room, questions about metal devices, answers patients questions, etc.) Did anyone from the radiology team provided any information to these patients? If so, did they do the same to groups 2 and 3? If so, was this a standardized procedure, i.e., all participants received some identical information from the hospital team?
  • Still in study design, between groups 2 e 3 some conditions vary may having some potential unknown impact on results: in G2, the information was given by a researcher and did not included equipment visualization; in group 3, the researcher showed the video, which was performed by a technician (not present in the previous condition) and equipment was shown (not present in the previous condition) The amount of information in G3 was larger (not quite the same). Taken together, these 2 questions make G3 information more complete and thus expected to reduce anxiety with more efficacy. Was this considered by the authors? Can you add any information that allows to understand that these 2 types of information were equal?
  • In population, the education level of patients is only presented as below / above secondary education, which is insufficient, as low education can condition reading and / or text understanding. It is likely that some people could not read or barely understand what was written; if this happened, how did authors control this situation, given the written instruction? Or patients with low educational level were included only in the visual group?
  • In Discussion, line 153, the authors mention that “Anxiety levels of participants during MRI were measured by State Anxiety Inventory”, but in fact, anxiety measures were taken before MRI: “MRI was performed within 20 minutes after measuring the state anxiety of the participants” (line 117). It’s not possible, then, to infer that with an interval of approximate 20 minutes waiting, the effect of movie watching /written information reading maintains the same effect than it was measured immediately after receiving this information. You would only be able to infer the effect of different types of information on MRI if you would have also measured anxiety immediately after MRI, and this is, in my point of view, an important limitation of your study. For this reason, statements of “anxiety reduction during MRI” (e.g. line 201) as a result of your study should be replaced by “anxiety reduction before MRI” and this limitation should be stated.
  • Revision should include a correction of the text by a native English-speaker.

Author Response

dear reviewer;

all revision suggestions were applied.

revisions are given in the attachment.

---

Response to Reviewer 2 Comments

 Point 1: Introduction:  Introduction In Introduction, sentence from line 46 to 50 about strategies to reduce patients anxiety should be supported by references, highlighting the importance of MRI previous information in anxiety, to support the present study  (line: 49-52) (line: 68-72)

Response 1: all these revision suggestions have been applied (line 49-72)

  • Point 2: Metods: In study design, a major issue which is not explained and may raise ethical questions is the control group. This group, according to good practices standards in MRI, should receive some information about the procedure before it starts. Usually, a technician or an assistant give some verbal information to the patient before the procedure (welcomes him, shows him the changing room, questions about metal devices, answers patients questions, etc.) Did anyone from the radiology team provided any information to these patients? If so, did they do the same to groups 2 and 3? If so, was this a standardized procedure, i.e., all participants received some identical information from the hospital team? ( line 135-137)
  • all these revision suggestions have been applied (wk 14)
  • Still in study design, between groups 2 e 3 some conditions vary may having some potential unknown impact on results: in G2, the information was given by a researcher and did not included equipment visualization; in group 3, the researcher showed the video, which was performed by a technician (not present in the previous condition) and equipment was shown (not present in the previous condition) The amount of information in G3 was larger (not quite the same). Taken together, these 2 questions make G3 information more complete and thus expected to reduce anxiety with more efficacy. Was this considered by the authors? Can you add any information that allows to understand that these 2 types of information were equal? (line: 134-160)
  • all these revision suggestions have been applied (2.3. Information method)
  • In population, the education level of patients is only presented as below / above secondary education, which is insufficient, as low education can condition reading and / or text understanding. It is likely that some people could not read or barely understand what was written; if this happened, how did authors control this situation, given the written instruction? Or patients with low educational level were included only in the visual group?
  • all these revision suggestions have been applied (table 1)

Point 3: Discussiın:  In Discussion, line 153, the authors mention that “Anxiety levels of participants during MRI were measured by State Anxiety Inventory”, but in fact, anxiety measures were taken before MRI: “MRI was performed within 20 minutes after measuring the state anxiety of the participants” (line 117). It’s not possible, then, to infer that with an interval of approximate 20 minutes waiting, the effect of movie watching /written information reading maintains the same effect than it was measured immediately after receiving this information. You would only be able to infer the effect of different types of information on MRI if you would have also measured anxiety immediately after MRI, and this is, in my point of view, an important limitation of your study. For this reason, statements of “anxiety reduction during MRI” (e.g. line 201) as a result of your study should be replaced by “anxiety reduction before MRI” and this limitation should be stated.

Response 3 : line 248-250 and box of C24

Reviewer 3 Report

Materials and methods

The inclusion criteria were not well described.

Pease explain what it means by, ‘people who have not been informed about MRI before’ (page 2, line 62)

The type of MRI examination is not presented. Is it MRI examination of head or abdominal parts?

The sentence in Page 2, line 62 can be deleted

The sample selection method is not presented. Please clarify the selecting process of study sample. 

Methods section is unorganized and does not provide detailed information to be able to replicate the study

Results.

Table 1 presents sociodemographic characteristics of patients. Why p values?

DISCUSSION.

The meaning and importance of the findings are not explained. The possible alternative reasons of the findings are not considered. In addition to comparing the study findings with those from the existing literature, I suggest that some hypotheses should be made in an attempt to explain them in their own context.

Conclusion: page 7, line 203-204. Is it a result of this study?

Acknowledgements. Please thank all participants

The manuscript needs a review of English language use (grammatical errors are noted).

Author Response

dear reviewer;

All revision suggestions have been applied and the final version of the article was reviewed by professional English editing service. 

---

Response to Reviewer 3 Comments

Point 1: Metods: Materials and methods

The inclusion criteria were not well described.:  (line 82-88)

Pease explain what it means by, ‘people who have not been informed about MRI before’ (page 2, line 62) (line 82-88)

The type of MRI examination is not presented. Is it MRI examination of head or abdominal parts? (line 82-88)

The sentence in Page 2, line 62 can be deleted (line 82-85) (wk 10 box)

The sample selection method is not presented. Please clarify the selecting process of study sample. (line 89-123: section:  2.2study sample)

Methods section is unorganized and does not provide detailed information to be able to replicate the study (line 175-173) ((2.1. Study design and population, 2.2. Sample size and selection 2.3. Data collection tools  2.4. Information method  2.5. Statistical analysis)

Point 2: Results:  Results.

Table 1 presents sociodemographic characteristics of patients. Why p values?  (When creating groups, it was paid attention that there was no difference between the groups demographically.)

Point 3: Discussion: DISCUSSION.

The meaning and importance of the findings are not explained. The possible alternative reasons of the findings are not considered. In addition to comparing the study findings with those from the existing literature, I suggest that some hypotheses should be made in an attempt to explain them in their own context. ( line 255-260)

Conclusion: page 7, line 203-204. Is it a result of this study? (line 263-265 : yes it is a result of study)

Acknowledgements. Please thank all participants  (line 276)

English editing:

Points: all reviewers: The manuscript needs a review of English language use (grammatical errors are noted).

Response: The final version of the article was reviewed by professional English editing service.

Round 2

Reviewer 2 Report

  • It was suggested that the sentence “In order to reduce anxiety during MRI, many approaches such as sedation, lying on your face instead of on your back, rehearsal with MRI simulator before MRI, relaxation exercises and information are recommended (lines 75 to 77 in the revised text) should be supported by literature references. That was not done by authors.
  • In “Information method” it is stated that “The same information used in the written brochure was explained by a technician in the visual video. In the video, the participants were shown the MR device, the sound and themagnetic resonance imaging” (lines 339-341). As I mentioned previously, in first revision, this makes the two conditions different, since the video adds more information to the patients and so, when we compare the results between these groups, we cannot say that they received quite the same information. The video group received more information (MR device visualization and MR device sound), and for this reason, one cannot say that the video is more effective in reducing anxiety than the written information, because we don’t know if this difference is influenced by seeing and hearing the equipment sound before MRI, or to the effect of seeing instead of reading, or both. This must be stated in limitations, providing a suggestion to make the two conditions more similar in the future.
  • Also, I strongly recommend, again, that the text should be revised by a native English-speaker.

Author Response

dear reviewer;

the revisions made are stated in cover letter reviewer report round 2

the revisions made can be seen in the attached file 

thanks for your contributions.

----

Review report raund 2:

Response to Reviewer 2 Comments:

Point 1:  It was suggested that the sentence “In order to reduce anxiety during MRI, many approaches such as sedation, lying on your face instead of on your back, rehearsal with MRI simulator before MRI, relaxation exercises and information are recommended (lines 75 to 77 in the revised text) should be supported by literature references. That was not done by authors.

Response 1:   8th and 9th references cited as a reference to this section. 9. reference was newly added and the all reference list in the entire text was rearranged

Point 2: In “Information method” it is stated that “The same information used in the written brochure was explained by a technician in the visual video. In the video, the participants were shown the MR device, the sound and themagnetic resonance imaging” (lines 339-341). As I mentioned previously, in first revision, this makes the two conditions different, since the video adds more information to the patients and so, when we compare the results between these groups, we cannot say that they received quite the same information. The video group received more information (MR device visualization and MR device sound), and for this reason, one cannot say that the video is more effective in reducing anxiety than the written information, because we don’t know if this difference is influenced by seeing and hearing the equipment sound before MRI, or to the effect of seeing instead of reading, or both. This must be stated in limitations, providing a suggestion to make the two conditions more similar in the future.

Response 2: When we say "written and visual information is the same", we want to emphasize that the contents of the two methods are the same. visual information differs from written information due to its audio and video content. however, it has been reported in the literature that both methods can reduce anxiety before MRI. Indeed, in our study, both methods were found to decrease anxiety before MRI. In our study we determined which method is more effective. Based on your recommendation, the subject is mentioned in the limitations section.